# Self- and informant-reported personality traits and vaccination against COVID-19

Kadri Arumäe[1], Anu Realo[1,2], Liisi Ausmees[1], Jüri Allik[1], Tõnu Esko[3], Krista Fischer[3,4], Uku Vainik[1,3,5], René Mõttus[1,6]*, Estonian Biobank Research Team[¶]

1 Institute of Psychology, University of Tartu, Tartu, Estonia, 2 Department of Psychology, University of Warwick, Coventry, England, 3 Institute of Genomics, University of Tartu, Tartu, Estonia, 4 Institute of Mathematics and Statistics, University of Tartu, Tartu, Estonia, 5 Montreal Neurological Institute, McGill University, Montreal, Canada, 6 Department of Psychology, University of Edinburgh, Edinburgh, United Kingdom

¶ The complete membership of the author group can be found in the Acknowledgments.
* rene.mottus@ed.ac.uk

**Data Availability Statement:** The procedures for requesting access to Estonian Biobank data are outlined at https://genomics.ut.ee/en/content/estonian-biobank.

## Abstract

As COVID-19 vaccines' accessibility has grown, so has the role of personal choice in vaccination, and not everybody is willing to vaccinate. Exploring personality traits' associations with vaccination could highlight some person-level drivers of, and barriers to, vaccination. We used self- and informant-ratings of the Five-Factor Model domains and their subtraits (a) measured approximately at the time of vaccination with the 100 Nuances of Personality (100NP) item pool ($N = 56,575$) and (b) measured on average ten years before the pandemic with the NEO Personality Inventory-3 (NEO-PI-3; $N = 3,168$). We tested individual domains' and either items' (in the 100NP sample) or facets' (in the NEO-PI-3 sample) associations with vaccination, as well as their collective ability to predict vaccination using elastic net models trained and tested in independent sample partitions. Although the NEO-PI-3 domains and facets did not predict vaccination ten years later, the domains correlated with vaccination in the 100NP sample, with vaccinated people scoring slightly higher on neuroticism and agreeableness and lower on openness, controlling for age, sex, and education. Collectively, the five domains predicted vaccination with an accuracy of $r = .08$. Associations were stronger at the item level. Vaccinated people were, on average, more science-minded, politically liberal, respectful of rules and authority, and anxious but less spiritual, religious, and self-assured. The 100NP items collectively predicted vaccination with $r = .31$ accuracy. We conclude that unvaccinated people may be a psychologically heterogeneous group and highlight some potential areas for action in vaccination campaigns.

## Introduction

Vaccination is a safe and effective way to prevent the incidence and reduce the severity of COVID-19 [1–3]. Besides reducing risks to individuals, vaccination helps to contain the virus' spread to others [4, 5], which also helps to prevent the emergence of new variants. But not everybody is willing to vaccinate, and the role of personal choice may only grow with the perils

**Funding:** This work has been funded by Estonian Research Council's (https://etag.ee/en/) personal research funding start-up grants PSG656 and PSG759, issued to UV, and Estonian Research Council's team grants PRG2190, issued to RM, and PRG1291, issued to TE. The funders had no role in study design, data collection and analysis, decision to publish, or preparation of the manuscript.

**Competing interests:** The authors have declared that no competing interests exist.

of the pandemic fading away and with vaccines' accessibility. So, understanding psychological reasons that either hinder or drive people to vaccinate can help facilitate reaching vaccination targets [6]. Because personality traits are typically conceptualized as enduring dispositions to feel, think, and behave consistently in similar situations, some of them may act as drivers of or barriers to vaccination, so knowing their associations with vaccination behavior could provide important insight into how vaccination programs should be organized and communicated to the public. To this end, we examined whether broad personality domains and narrower nuances can predict vaccination against COVID-19 both concurrently and many years in the future.

## Vaccination and the personality trait hierarchy

Personality traits are organized hierarchically, with broad domains like the Big Five splitting into narrower traits like facets and micro-traits, also called nuances [7]. Whereas previous research on personality traits and vaccination (reviewed below) has primarily focused on broader traits, narrower traits likely have additional utility in helping to predict and understand vaccination decisions. Specifically, because traits grouped into the same domain often have different associations with outcomes of interest [8–10], narrower traits can offer more information. For instance, within the neuroticism domain, vaccination may plausibly correlate differently with anxiety (which could motivate vaccination to avoid getting infected, thus correlating positively with vaccination) and angry hostility (which could diminish one's concern for getting ill and spreading the disease, thus correlating negatively with vaccination). Meaningful within-domain differences may even exist when domains appear unrelated to vaccination. For example, although extraversion may not have a consistent domain-level link to vaccination (as reviewed below), this may be because some extraverts doubt the risk presented by the virus [11] and deem vaccination unnecessary, while others may prefer to vaccinate to attend social events. If so, important drivers of vaccination may remain unknown unless associations with lower-level traits are also examined.

Exploring associations with domains' subcomponents—facets—is common in personality research. However, an even more detailed description of personality trait–vaccination correlations could consider personality nuances [12]. Often indexed by individual questionnaire items, these traits are narrower than facets, yet have many properties similar to domains and facets, including rank-order stability, heritability, and cross-rater agreement [13, 14]. Nuances may have no descriptive advantage over domains or facets in relation to some COVID-19 mitigation behaviors [15], but they typically do carry additional information about a range of real-world outcomes, sometimes even entirely driving these outcomes' associations with personality traits [16–18]. Where this applies, outcomes could be (a) weakly linked to many narrow personality traits rather than a few broad traits such as personality domains or (b) strongly linked to just a few narrow traits described with personality nuances. Either could be true for vaccination decisions. Although it may be laborious to interpret potentially hundreds of nuances' correlations with an outcome (such as vaccination), the possible benefits of nuanced descriptions include understanding some of the most important and possibly actionable drivers and barriers of vaccination. To clarify which level of resolution—domains, facets, or nuances—captures the most information about vaccination, we can systematically compare their out-of-sample predictive accuracies in sufficiently large samples [7].

## Personality traits and vaccination against COVID-19: Prior evidence

Cross-sectionally, agreeableness may track pro-vaccination attitudes [19], vaccination willingness [20], and lower vaccine hesitancy and resistance [21] while two narrower traits within

that domain, prosociality and altruism, have been related to vaccination intention [22] and lower vaccine hesitancy [21], respectively. The openness domain has been related to vaccination willingness [23], consistently with pro-vaccination stances tracking political liberalism [19, 22, 24, 25]. Neuroticism may be inversely related to vaccination, being positively related to vaccine hesitancy [21] and negatively to pro-vaccination attitudes [19] and self-reported vaccination [24]. Conscientiousness has been positively associated with pro-vaccination attitudes [19], vaccination intention [26], lower vaccine hesitancy and resistance [21], and self-reported vaccination behaviour [24], but also to vaccine hesitancy due to unknown future health risks [11, 27]. Extraversion has been linked to both vaccination intention [26] and, to the contrary, anti-vaccination attitudes, at least in people aged 50–65 [24]. Longitudinal studies have linked vaccine hesitancy to past measurements of neuroticism but no other Big Five domain [11] and vaccine intention to pre-pandemic impulsivity and psychopathy but not altruism [28]. From the pre-pandemic variations of the Big Five, only higher openness predicted vaccination rates across 48 US states [29], whereas higher pre-pandemic neuroticism predicted vaccination rates across 56 countries [30].

To summarize, the existing evidence on the associations between personality traits and vaccination is somewhat inconsistent: For some traits (e.g., altruism, neuroticism), results based on cross-sectional studies differ from those obtained with longitudinal studies, and some traits (e.g., conscientiousness, extraversion) conflicting results have been obtained. With most studies focusing either on the Big Five traits or a small number of traits hypothesized to be linked to vaccination, there has not been a systematic attempt to provide a detailed account of traits' links to vaccination against COVID-19.

## Group differences in personality trait–vaccination associations

While some personality trait–vaccination relations may turn out to generalize across individuals' demographic backgrounds, other traits could be more relevant for vaccination decisions in some groups than others. There is already evidence that personality traits' associations with vaccination willingness may depend on age and sex. For instance, traits reflecting risk aversion and self-interest could drive older people to vaccinate, while prosocial traits could be more relevant for young people, particularly men [20]. If such moderation effects of age, sex, or other demographic variables exist, knowing them could provide further information about the specific drivers of and hindrances to vaccination, potentially allowing for more targeted interventions. For instance, if more anxious older people are apprehensive of vaccines, then campaigns, especially those targeted at this demographic group, could focus on vaccines' safety or risk-benefit trade-offs; if younger men's decision to vaccinate is linked to prosociality, campaigns could appeal to altruistic motives, especially if targeted at this demographic group. It would be especially valuable to understand what personality traits are most strongly related to vaccination in groups with the lowest vaccination uptake because effectively motivating these people could have the highest impact on overall vaccination rates. Vaccination campaigns could thus prioritize messages most likely to be effective in subgroups with the lowest vaccine uptake.

## Moving beyond self-reports

Self-reports provide an efficient and scalable method for assessing many traits and outcomes, including personality traits as well as vaccination attitudes and behavior. However, any single assessment method is subject to biases [31]. For example, people can provide biased assessments of their personality traits when unaware of their typical behaviors, have more information on their past behaviors than they can accurately summarize as trait ratings, or misreport

their traits to maintain their self-perceptions [32]. Self-reports of vaccination behavior can also be biased: If vaccination is seen as socially desirable, some unvaccinated people may report being vaccinated [33]. If so, single-method biases can deflate or inflate personality trait-vaccination correlations.

Single-method biases can be mitigated by incorporating multiple assessment methods. For vaccination, information would ideally be obtained from objective sources such as official vaccination records. For personality ratings, the most viable option is to use participants' knowledgeable informants to rate their traits, as informants typically do not have the same biases but provide different perspectives on the target's personality. In particular, numerous researchers have called for multi-method or multi-source approaches to measuring personality [31, 32]. While people themselves may be more accurate in rating some of their traits (e.g., those not easily observed), informants' ratings can be more accurate for others (e.g., more evaluative traits [34]), but agreement between independent raters can lend credibility to the results. Unfortunately, multi-rater studies are still rare, especially with sufficiently large samples to provide robust results. For vaccination against COVID-19 specifically, we know of no studies that have tested associations between individual-level personality traits and objective vaccination status.

## The present study

With a set of preregistered analyses (https://osf.io/vwjmk), we investigated the associations between the self- and informant-rated personality traits and vaccination status, timing, and the number of doses based on official medical records. We used two large, partly overlapping population-based samples from Estonia: one where personality traits were assessed at approximately the same time as the population was undergoing vaccination against COVID-19 during the pandemic, and the other where the Five Factor Model personality traits had been assessed ten years before the pandemic on average.

In the first dataset, personality traits were measured with an item pool designed to cover a large but mostly non-redundant collection of narrow personality traits alongside the Five-Factor Model (FFM) and HEXACO domains: the 100NP ("The 100 Nuances of Personality" [35]). With broad coverage of traits, these data were particularly well-suited to assess broad and narrow personality traits' overall predictive power for vaccination status and to identify specific traits with the strongest links to it. To identify the trait hierarchy level most suitable for describing vaccination's personality correlates, we compared the domains' and their nuances' accuracies in predicting vaccination. We also explored whether personality traits' relevance to vaccination varied with age, sex, education, occupation, or residency. To identify potentially most vaccination-relevant personality traits, we tested domains' and nuances' associations with vaccination status in the whole sample and specifically in the demographic groups with the lowest vaccination rates. In the second dataset, we tested how strongly the personality domains, facets, and nuances, as measured with the NEO Personality Inventory (NEO-PI-3 [36]), could predict future vaccination status and which domains and facets were most predictive.

We primarily focused on vaccination status as a binary outcome (vaccinated versus unvaccinated). However, vaccines' efficacy depends on the number of doses: Not only are additional doses often necessary to achieve maximal protection, but vaccines' immunizing effect also wears off with time as antibody levels decline [37]. Vaccination timing also matters, as earlier vaccination helps to limit the pandemic's spread, and it is possible that vaccines' early and late adopters differ psychologically. Thus, we additionally examined personality traits' predictive accuracy for the number of doses received and the first dose's timing (early or late).

## Method

### Ethics

The study was approved by the Estonian Committee on Bioethics and Human Research (1.1-12/1515, 12.04.2022). All participants of both samples provided written informed consent with their signature either digitally or on paper. The authors had no access to information that could identify individual participants during or after data collection.

### Participants

We used two samples from the Estonian Biobank (EB) of the Estonian Genome Center, University of Tartu, a cohort of over 200,000 Estonian residents (about 20% of the Estonian adult population) recruited by medical personnel and media campaigns throughout the country [38]. Participants provided DNA samples alongside metabolomic, various demographic and health data. We tested concurrent personality–vaccination associations in a subsample of EB participants whose personality traits had been measured between November 2021 and April 2022 with a diverse pool of 198 items (the 100NP). These participants were contacted via email (followed by up to two reminders, if and as necessary), and the study was also introduced to the public through newspapers, radio, television, and social media. In return for participation, participants were offered feedback on their personality domain scores. Participants could opt to complete the survey in either Estonian or Russian. After completing the survey, participants were asked for the email address of an acquaintance (e.g., spouse, partner, relative, or friend) who would complete the 100NP's informant-report forms about the participants. The sample is described in more detail elsewhere [39].

The initial sample with complete self-reported personality data included 77,281 participants. We excluded participants with missing information on age, sex, or education and those under 18 years old. We further removed participants who had more than 20% genetic relatedness with each other as closely related participants in the sample inflate predictor–outcome associations and affect predictive accuracy [40]. The exclusion algorithm preferred participants with lower degrees of relatedness to others, as well as unvaccinated participants (as unvaccinated people were less represented). Relatedness was determined with PLINK 1.9 [41, 42] command PI_HAT. After these exclusions, complete self-report personality data were available for 56,575 adults aged 18 to 96 years, with informant reports available for 15,244 of them.

We further tested across-time associations in an EB subsample where personality traits had been measured with the NEO-PI-3 [36] between 2008 and 2017—that is, between 4 and 14 years before vaccination against COVID-19 began (63% had completed the NEO-PI-3 by the end of 2011 and 93% by the end of 2013). Of the initial sample of 3,601 people with complete self-report personality ratings, information on age, sex, and education was available for 3,168 participants between ages 19 and 91 who were not closely related to each other and who had not died by the end of 2021 (by which time people had had the opportunity to get vaccinated against COVID-19 for the first time). Participants were also asked to find an informant to complete the informant form of the NEO-PI-3; complete informant-report data were available for 3,168 participants. Because the inventory was in Estonian, most participants likely belonged to the ethnic majority group, Estonians. An overview of the data collection procedures has been given previously [38].

As 1,273 people had responded to both inventories, the two samples partly overlapped. Both samples' characteristics are detailed in Table 1. Details on vaccination stage, timing, and vaccine types are reported in S1 Table, and informants' characteristics and relations to the targets are in S2 Table of the supplementary document. Compared to the Estonian population

**Table 1. Sample characteristics.**

| Characteristic | 100NP data | | | NEO-PI-3 data | | |
|---|---|---|---|---|---|---|
| | All participants (*N* = 56,575) | Vaccinated (*n* = 49,292) | Unvaccinated (*n* = 7,283) | All participants (*N* = 3,168) | Vaccinated (*n* = 2,718) | Unvaccinated (*n* = 450) |
| Age: mean (*SD*) | 47.60 (14.20) | 48.14 (14.33) | 43.93 (12.73) | 55.25 (15.85) | 56.07 (15.88) | 50.31 (14.75) |
| Age group | | | | | | |
| Younger (< 50 years) | 24,284 (42.92%) | 21,991 (44.61%) | 2,293 (31.48%) | 1,217 (38.42%) | 1,064 (39.15%) | 153 (34.00%) |
| Older (≥ 50 years) | 32,291 (57.08%) | 27,301 (55.39%) | 4,990 (68.52%) | 1,951 (61.58%) | 1,654 (60.85%) | 297 (66.00%) |
| Sex | | | | | | |
| Male | 16,687 (29.50%) | 14,946 (30.32%) | 1,741 (23.90%) | 1,858 (58.64%) | 1,651 (60.73%) | 207 (46.00%) |
| Female | 39,888 (70.50%) | 34,346 (69.68%) | 5,542 (76.10%) | 1,310 (41.36%) | 1,067 (39.27%) | 243 (54.00%) |
| Education | | | | | | |
| With higher education | 32,812 (58.00%) | 29,653 (60.16%) | 3,159 (43.37%) | 1,311 (41.38%) | 1,177 (43.30%) | 134 (29.78%) |
| Without higher education | 23,763 (42.00%) | 19,639 (39.84%) | 4,124 (56.63%) | 1,857 (58.62%) | 1,541 (56.70%) | 316 (70.22%) |
| Employment | | | | | | |
| Blue-collar | 8,557 (15.58%) | 6,764 (14.11%) | 1,793 (25.69%) | – | – | – |
| White-collar | 46,365 (84.42%) | 41,179 (85.89%) | 5,186 (74.31%) | – | – | – |
| Residency | | | | | | |
| Rural | 14,901 (27.16%) | 12,926 (26.58%) | 1,975 (31.65%) | – | – | – |
| Urban | 39,963 (72.84%) | 35,697 (73.42%) | 4,266 (68.35%) | – | – | – |
| Language | | | | | | |
| Estonian | 53,166 (93.97%) | 46,456 (94.25%) | 6,710 (92.13%) | 3,168 (100%) | 2,718 (100%) | 450 (100%) |
| Russian | 3,409 (6.03%) | 2,836 (5.75%) | 573 (7.87%) | 0 (0.00%) | 0 (0.00%) | 0 (0.00%) |

Counts and percentages per class are presented unless specified otherwise. For employment and residency, numbers do not add up to the total sample sizes due to missing data. Participants' age is given at the time vaccines became available to the general population of Estonia (May 2021).

(https://andmed.stat.ee/et/stat/rahvastik__rahvastikunaitajad-ja-koosseis__rahvaarv-ja-rahvastiku-koosseis), women, people with higher education, and vaccinated people were over-represented in both samples. The participants' mean age at the time vaccines became available exceeded the Estonian average slightly in the 100NP dataset and considerably in the NEO-PI-3 dataset. The proportions of people living in urban and rural areas (available for the concurrent sample) roughly corresponded to the Estonian population.

## Materials

**100NP.** The 100NP is a pool of 198 items designed to cover the content of broad domains like those of the FFM and HEXACO and assess narrow personality traits (nuances) within and beyond them, prioritizing individual items' high reliability and low redundancy to maximize overall information capture; a detailed account of its development and properties is available at [35] and the rationale for such a scale development model is presented in [43]. Most items were drawn from the International Personality Item Pool [44] or the Synthetic Aperture Personality Assessment item pool [45] by iteratively dropping items with lower variance from highly correlating item pairs and testing the remaining items for test-retest reliability, limited redundancy, and coverage of widely used questionnaires like the NEO-PI-3 and HEXACO [46]. In this multi-step process, items with lower reliability and high redundancy were dropped, testing alternative items instead. Although selected for broad coverage and hence little overlap, some trait content was represented with two highly similar items (e.g., "Am always worried about something" and "Rarely worry") to allow testing for inconsistent and acquiescent responding, for example. Answers were provided on a scale from 1 (*completely*

*inaccurate*) to 6 (*completely accurate*). The 100NP was developed in English and the items were then translated into Estonian and Russian, using independent back-translations followed by numerous rounds of revisions and consultations involving both the authors of the present paper and numerous English-Estonian, English-Russian or Estonian-Russian bilinguals. Self- and observer-report versions of the 100NP were used in the present study. The 100NP has been used to examine personality traits' associations with life satisfaction [47] as well as the personality profiles of a broad range of occupations [48].

We treated individual items as markers for personality nuances. For domain-level analyses, the items were assigned to five domains using principal component analysis (PCA) in self-report data using oblimin rotation. Each item was assigned to one of the five domains if it had a loading of at least |.40| on the respective component. Thirty-one items were assigned to the neuroticism domain (McDonald's ω = .95), 28 to agreeableness (ω = .92), 18 to extraversion (ω = .88), 17 to conscientiousness (ω = .88), and 12 to openness (ω = .84) with the remaining 92 items not assigned to any domains. The domains' intercorrelations did not exceed |$r$| = .36 (the highest correlation being between openness and extraversion), which, desirably, is considerably lower than domains' inter-correlations in many other Big Five instruments [49]. The items assigned to each domain are listed in S3 Table.

**NEO-PI-3.** The Estonian version of the NEO-PI-3 is a slightly modified version of the Estonian NEO-PI-R [50]. This 240-item inventory measures the domains of the FFM and their 30 facets. The items were answered on a scale from 0 (*strongly disagree*) to 4 (*strongly agree*). Participants completed the inventory's self-report form; informants completed its observer-report form.

Because 1,273 people had completed both 100NP and NEO-PI-3, it was also possible to assess each FFM domain's content overlap in the two inventories. The personality domains' zero-order correlations in the two datasets were .70 for neuroticism, .65 for extraversion, .60 for openness, .61 for agreeableness, and .63 for conscientiousness. Because the participants completed the two questionnaires 10.37 years apart on average (range 4.34–13.57) and typical rank-order stability over several years is around .70 [51], these correlations indicate the high level of construct validity of the two instruments.

**Vaccination.** We obtained vaccination data from electronic medical records through a centralized health information database (The Health and Welfare Information Systems Centre; www.tehik.ee). Three dependent variables were extracted for each participant. First, vaccination status (a binary variable: vaccinated, having received at least one dose of a COVID-19 vaccine, versus unvaccinated, having received no COVID-19 vaccine by May 2, 2022). Second, vaccination stage (an ordinal variable with four levels: unvaccinated; primary vaccination series started but not completed; primary series completed; at least one booster dose received). The primary vaccination series consisted of two doses for all but the Jcovden (Janssen) vaccine for which the primary series consisted of one dose. People who had received one dose of the Janssen vaccine were thus considered fully vaccinated (primary series completed). Third, vaccination timing was (a binary variable: early, having received a vaccine dose by August 31, 2021, versus late, having received the first dose on September 1, 2021 or later). In analyses with vaccination timing as the dependent variable, we excluded unvaccinated and priority-vaccinated participants (i.e., those vaccinated before May 17, 2021 when vaccines became available to the general population in Estonia).

**Moderators.** We considered age, sex, the highest level of education, residency, and occupation as variables moderating personality trait–vaccination associations in the 100NP dataset. Education was measured as an ordinal variable with nine levels ranging from no education to PhD. For residency, participants were asked to indicate if they lived in an urban (large, medium, or small city) or rural area (other types of settlements). Occupation was measured

with the International Classification of Occupations ([http://www.ilo.org/public/english/bureau/stat/isco/index.htm](http://www.ilo.org/public/english/bureau/stat/isco/index.htm)). For moderation analyses, the non-binary moderators of interest were dichotomized into groups of younger and older age (under 50 years / at least 50 years of age on May 17, 2021), higher and lower education (with higher education, i.e., at least a completed bachelor's degree or equivalent / without higher education), urban and rural residency, and white-collar and blue-collar occupation (with managers, professionals, technicians and associate professionals, clerical support workers, and service and sales workers classified as white-collar and craft and related trades workers, plant and machine operators and assemblers, elementary occupations, and armed forces occupations classified as blue-collar).

## Statistical analyses

**Concurrent multivariable associations.**   We assessed self- and informant-reported domains' and items' predictive power for the three outcomes—vaccination status, stage, and timing—with elastic net models. Elastic net regression [52] maximizes prediction by selecting and weighting predictors based on their importance in the model (i.e., in predicting the outcome) while avoiding inflated coefficients due to sampling idiosyncracies, thus enabling improved out-of-sample predictive performance compared to more traditional methods like multiple regression that tend to capitalize on samples' idiosyncracies that leads to lower out-of-sample prediction accuracy, especially with large predictor sets [7, 53]. It is crucial that the models are trained and tested in separate sets of data to completely avoid inflated prediction accuracy due to sampling error. Besides counteracting overfitting, this ensures that the differences between the domain and item models' predictive accuracies are not due to the larger number of predictors in the item models. We trained the models in a random 75% of the sample with 10-fold cross-validation and applied the trained models to the remaining 25% of the sample to test the concurrence/correlation between the outcomes' predicted and observed values (i.e., the models' predictive accuracy). We residualized items and domains for age, sex, and education. In model training, we used the binomial link for vaccination status and timing (binary variables) and a Gaussian link for vaccination stage (ordinal variable). For model testing, i.e., assessing predictive accuracy, we calculated Pearson's correlation between the predicted and observed vaccination stage but used classification error for the other two outcomes —that is, how often the model incorrectly classified people as vaccinated or unvaccinated (for the vaccination status dependent variable) or as vaccinated early or late (for the vaccination timing dependent variable), using two-tailed binomial tests to compare the models' performance to chance-level accuracy of 50%. To be able to compare prediction accuracies across the different types of dependent variables, we also used the binary outcomes' predicted and actual values to calculate Matthew's correlation coefficients (interpretable analogously with Pearson's correlation [54]).

Vaccinated people and especially those vaccinated early were considerably overrepresented in our samples compared to the Estonian population. Because such unequal distributions can bias predictions toward the majority groups in elastic net models (e.g., if 90% of the sample is vaccinated, the model will classify all people as vaccinated to obtain 90% accuracy), we applied random undersampling to balance the number of observations in the minority (unvaccinated and vaccinated late) and majority groups (vaccinated and vaccinated early). For the minority group, we used data from all participants; for the majority group, we selected observations randomly in equal numbers to the majority classes. Thus, for elastic net analyses involving the binary dependent variables, the final sample size in testing and training the models was twice the number of observations in the minority classes. We used the full sample for analyses with stage of vaccination.

**Comparing personality traits' and demographic variables' prediction accuracies (not preregistered).** To contextualize personality traits' links with vaccination, we compared their accuracy in predicting vaccination to that of demographic variables. Using a 75%/25% sample split among the participants with 100NP data, we trained three new elastic net models with binary vaccination status as the dependent variable and either (1) self-reported personality items, (2) five demographic variables (age, sex, education, occupation, and residency), or (3) the personality items (here not residualized for the demographic variables) and the demographic variables together as the predictors. These analyses allowed us to assess the incremental predictive accuracy of personality traits over demographic variables and vice versa. Again, we used random undersampling to balance the numbers of vaccinated and unvaccinated people. Because prediction accuracy can depend on sample size, we additionally limited the sample to cases with no missing data on any of the variables to ensure equal sample sizes for all three models.

**Moderation by demographic variables.** We tested whether any of the five demographic variables—age, sex, education, occupation, or residency—moderated personality traits' associations with vaccination status, using only the 100NP data for sample size considerations. Given our objective to compare moderation effects between vaccination and personality traits *altogether* (as opposed to each trait individually), we opted for an approach where we compared traits' predictive accuracy across the moderators' levels [7]. We residualized the personality variables (domains or items) for age, sex, and education, excluding the variable whose moderating effect was being tested. We trained elastic net models with 10-fold cross-validation among participants with one moderator level and tested them among participants with the other. This procedure enabled testing whether personality traits predicted vaccination stronger for one level of each moderator than the other, thereby indicating whether traits' associations with vaccination were moderated by any of the five demographic variables. The models were trained on younger people, men, people with higher education, people living in urban residential areas, and white-collar workers. For instance, for sex, the personality variables were residualized for age and education and the model was trained on males and tested on females. We used random under sampling for both the dependent and moderator variables, ensuring equal sample sizes in the training and testing data.

In the training samples, we obtained the cross-validated classification error using the *cv. glmnet()* function of the *glmnet* package [55]. In the testing samples, we estimated classification error as the proportion of errors made when applying the model fit in the training data. Moderation was tested by comparing the training and testing samples' classification errors with two-proportions tests (two-tailed).

**Associations with individual personality traits.** We tested individual personality traits' associations with vaccination status by using binomial logistic regression models with binary vaccination status as the outcome and a personality trait (domain, facet, or item) as the predictor, controlling for age, sex, and education. For items, we reported the 20 strongest item–vaccination associations in both self- and informant-ratings. Where demographic characteristics moderated the personality–vaccination associations in 100NP data, we also tested domains' and items' associations with vaccination status in the groups with lower vaccination rates. Again, the models accounted for age, sex, and education, excluding the moderating variable.

All analyses were conducted using R (version 3.6.3) in RStudio. We fit elastic net models using the *glmnet* package [55] with the mixing hyperparameter α set to .50. The regularization penalty hyperparameter λ's value was chosen to minimize cross-validation error. We applied false discovery rate (FDR) correction across analyses of one type involving the same dependent variable and same level of personality measurement (domains or items) separately for models using self- and informant-reported data.

**Across-time associations.** In the NEO-PI-3 dataset, vaccination was predicted respectively from all personality domains, facets, or items using the elastic net regression, and then from each domain and facet of the NEO-PI-3 with logistic regression. Both analyses were analogous to the multivariable analyses described above; both self- and informant-ratings were used. Again, we applied undersampling to the data in analyses with binary outcomes. Age, sex, and education were controlled for.

## Transparency and openness

We preregistered the analyses at Open Science Framework (https://osf.io/vwjmk). Exceptions are noted under *Statistical Analyses*. Analytic procedures were preregistered prior to accessing the data (except for the NEO-PI-3 data which have been used in previous studies). Analysis code is available at https://osf.io/qjs7e/. We report how we determined our sample size, all data exclusions, all manipulations, and all measures in the study. Due to data sharing limitations, vaccination, personality, or other health data cannot be publicly shared. The procedures for requesting access to Estonian Biobank data are outlined at https://genomics.ut.ee/en/content/estonian-biobank.

## Results

### Concurrent associations

**Multivariable associations.** Domains' and items' accuracies in predicting vaccination are reported in Fig 1 and detailed in S4 Table. The five domains' predictive accuracy was highest for binary vaccination status and vaccination timing (54% for both dependent variables, corresponding to an error rate of .46 or $r = .08$; informant-rated domains). Items' predictive

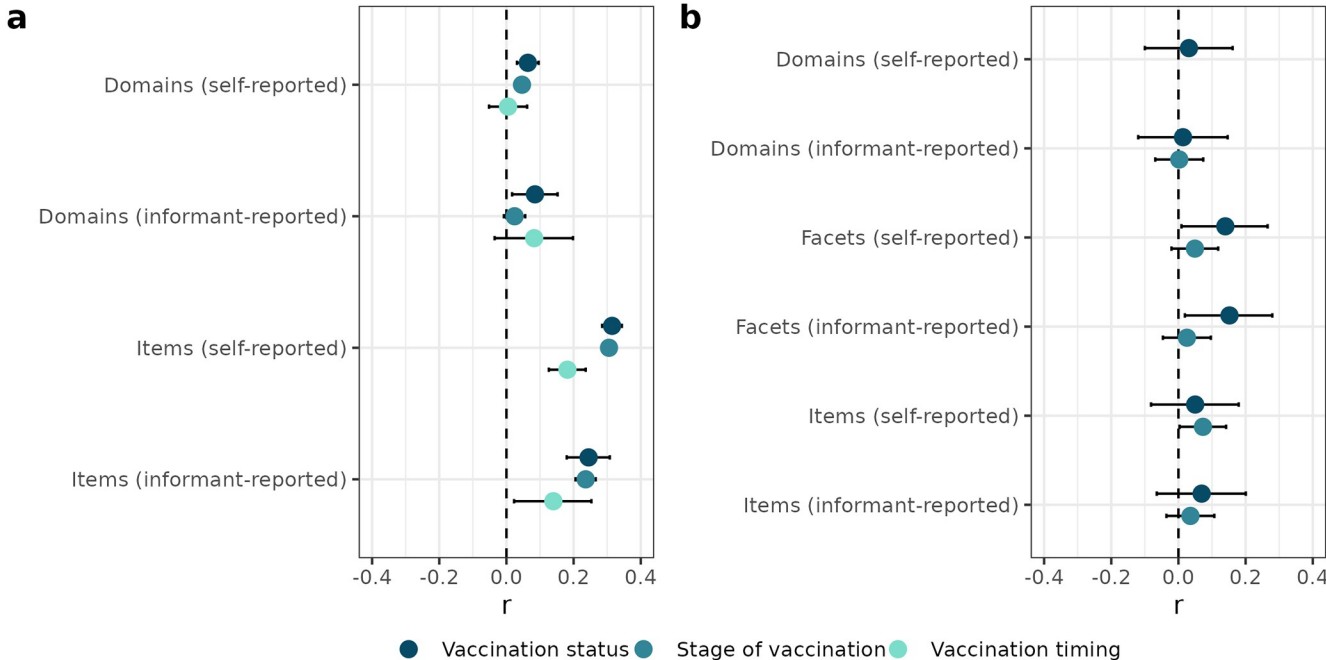

**Fig 1. Predicting vaccination from personality traits.** Trait–vaccination associations in the 100NP sample (panel *a*) and the NEO-PI-3 sample (panel *b*). The associations are reported in detail in S4 and S5 Tables. The domains of the NEO-PI-3 were unable to predict stage of vaccination (elastic net weights were zero for all domains). Prediction of vaccination timing in the NEO-PI-3 dataset is not reported because sample sizes were too low (i.e., no more than 130 people in training and 44 people in testing data) to enable meaningful prediction of this dependent variable in the NEO-PI-3 data.

accuracy was up to 66% (*r* = .31) for binary vaccination status, *r* = .31 for stage of vaccination, and 59% (*r* = .18) for vaccination timing. In most cases, self-ratings had higher predictive accuracies than informants' ratings. We emphasize that separating model training and testing ensured that items' higher predictive accuracy was not because of the models' higher complexity (i.e., overfitting).

**Comparing personality traits' and demographic variables' prediction accuracies.** Without adjusting for any covariates, the 198 self-reported items collectively predicted vaccination status with higher accuracy (71%; *r* = .42) than the demographic variables (62%; *r* = .25), suggesting that detailed personality measurements contain more information relevant to vaccination than a collection of demographic characteristics. Adding demographic variables to personality items marginally increased prediction accuracy to 72% (*r* = .43), suggesting that the personality items can capture most of the vaccination-relevant information in the demographic variables but not the other way around.

The error rates and correlations are further detailed in S6 Table; the items' weights in the item models are presented in S7 Table along with their weights after residualizing for age, sex, and education. Notably, 122 of the 198 items (62%) had non-zero weights for predicting vaccination status when not residualized for any covariates and 61 (31%) had non-zero weights after residualizing for the covariates, suggesting that many narrow traits are uniquely linked with and hence contribute towards the prediction of vaccination.

**Moderation by demographic characteristics.** Each of the five demographic variables statistically significantly moderated the personality–vaccination link (Table 2), but the effects were appreciably larger for education and occupation (the respective differences in classification accuracies were up to .18 and .25) than for age, sex, and residency. Thus, we selected the education and occupation groups with lower vaccination coverage to identify traits that were especially strongly related to vaccination, besides the whole sample-level analyses of these associations. Vaccination rates were lower among people without higher education (83%) than people with higher education (90%) and among people with blue-collar jobs (79%) than people with white-collar jobs (89%).

**Trait-level associations in the whole sample.** For the 100NP domains, higher odds of vaccination were associated with lower self- and informant-rated openness, higher self-reported neuroticism, and higher informant-rated agreeableness (Fig 2).

Self- and informant-rated personality items were associated with vaccination status similarly: The correlation between the profiles of odds ratios based on the two sources of personality ratings was *r*(196) = .87, *p* < .001. The 20 strongest associations between vaccination status and self- and informant-rated personality items are shown in S10 Table; Fig 3 displays the items whose associations with vaccination were among the 20 strongest according to both sources of personality ratings. On average, vaccinated people were rated as more scientifically minded ("Believe that all events can be explained scientifically," "Am interested in science") while unvaccinated people were more spiritual and fatalistic ("Am a spiritual person," "Believe in the power of fate"). Also, vaccinated people were more likely to worry and feel self-conscious, stressed, and misunderstood ("Worry about what people think of me," "Am always worried about something," "Often feel that others misunderstand me," "Need reassurance") while unvaccinated people tended to be slightly more confident ("Don't hesitate to express an unpopular opinion," "Take risks") and have a higher opinion of themselves ("Am good at many things," "Consider myself good-looking").

Although many items with the strongest relations to vaccination could be associated with the openness domain, some of these associations differed in direction. Vaccinated people were more likely politically liberal ("Support liberal political candidates," "Don't consider myself religious," "Believe that everyone should have a say," "Treat all ethnicities and religions equally"), yet more conventional ("Like to be viewed as proper and conventional"), less

**Table 2. Demographic variables as moderators in the associations between personality traits and vaccination status.**

| Source | Predictors | Error rate (training) | Error rate (testing) | Difference [95% CI] | $\chi^{2\ a}$ | p | n (training) | n (testing) |
|---|---|---|---|---|---|---|---|---|
| Moderator: age | | | | | | | | |
| Self | Domains | .44 | .59 | -.15 [-.17; -.13] | 243.22 | < .001 | 5,546 | 5,546 |
| Self | Items | .35 | .40 | -.05 [-.07; -.04] | 34.40 | < .001 | 5,546 | 5,546 |
| Informant | Domains | .46 | .58 | -.12 [-.16; -.07] | 29.03 | < .001 | 1,061 | 1,061 |
| Informant | Items | .39 | .38 | .02 [-.03; .06] | 0.51 | .475 | 1,061 | 1,061 |
| Moderator: sex | | | | | | | | |
| Self | Domains | .42 | .49 | -.07 [-.09; -.05] | 39.88 | < .001 | 3,935 | 3,935 |
| Self | Items | .34 | .38 | -.04 [-.06; -.01] | 10.50 | .002 | 3,935 | 3,935 |
| Informant | Domains | .41 | .52 | -.11 [-.15; -.06] | 22.11 | < .001 | 945 | 945 |
| Informant | Items | .37 | .42 | -.05 [-.10; -.01] | 5.10 | .024 | 945 | 945 |
| Moderator: education | | | | | | | | |
| Self | Domains | .41 | .57 | -.15 [-.17; -.14] | 331.28 | < .001 | 7,031 | 7,031 |
| Self | Items | .31 | .44 | -.13 [-.15; -.11] | 252.28 | < .001 | 7,031 | 7,031 |
| Informant | Domains | .43 | .61 | -.18 [-.22; -.14] | 80.13 | < .001 | 1,265 | 1,265 |
| Informant | Items | .35 | .46 | -.11 [-.15; -.07] | 32.60 | < .001 | 1,265 | 1,265 |
| Moderator: residency | | | | | | | | |
| Self | Domains | .47 | .49 | -.02 [-.04; .00] | 2.52 | .224 | 3,672 | 3,672 |
| Self | Items | .34 | .36 | -.02 [-.04; .00] | 2.53 | .224 | 3,672 | 3,672 |
| Informant | Domains | .51 | .53 | -.01 [-.06; .04] | 0.20 | .655 | 813 | 813 |
| Informant | Items | .35 | .37 | -.02 [-.07; .03] | 0.52 | .625 | 813 | 813 |
| Moderator: occupation | | | | | | | | |
| Self | Domains | .46 | .54 | -.08 [-.10; -.05] | 31.75 | < .001 | 2,778 | 2,778 |
| Self | Items | .35 | .45 | -.10 [-.12; -.07] | 54.55 | < .001 | 2,778 | 2,778 |
| Informant | Domains | .44 | .69 | -.25 [-.31; -.19] | 59.64 | < .001 | 475 | 475 |
| Informant | Items | .41 | .49 | -.08 [-.15; -.01] | 5.82 | .016 | 475 | 475 |

For the moderators age, sex, education, residency, and occupation, testing data consisted of either participants who were younger, male, had higher education, lived in urban areas, or had white-collar occupations, respectively; training data consisted of participants who were older, female, had no higher education, lived in rural areas, or held blue-collar occupations, respectively.

[a] $df$ = 1.

eccentric ("Am considered to be kind of eccentric"), and more respectful of rules and authority ("Respect authority") compared to unvaccinated people ("Rebel against authority" and "Know how to get around the rules"). Unvaccinated people also tended to show a stronger preference for variety ("Dislike routine," "Adapt easily to new situations"). Thus, vaccination was positively related to political liberalism but negatively related to various other openness-related traits (unconventionality, rebelliousness, variety-seeking). Such within-domain nuanced ness in how personality traits are linked with vaccination explains why items were more predictive of vaccination outcomes than domains.

The domain- and item-level associations were broadly similar among people without higher education and people in blue-collar occupations. The differences from the complete sample are outlined in the supplementary document and the traits' associations in these subgroups shown in S9, S11, and S12 Tables.

### Across-time associations

The NEO-PI-3 domains and facets, measured approximately ten years before the beginning of the pandemic, did not significantly predict either vaccination status or stage (Fig 1). Similarly,

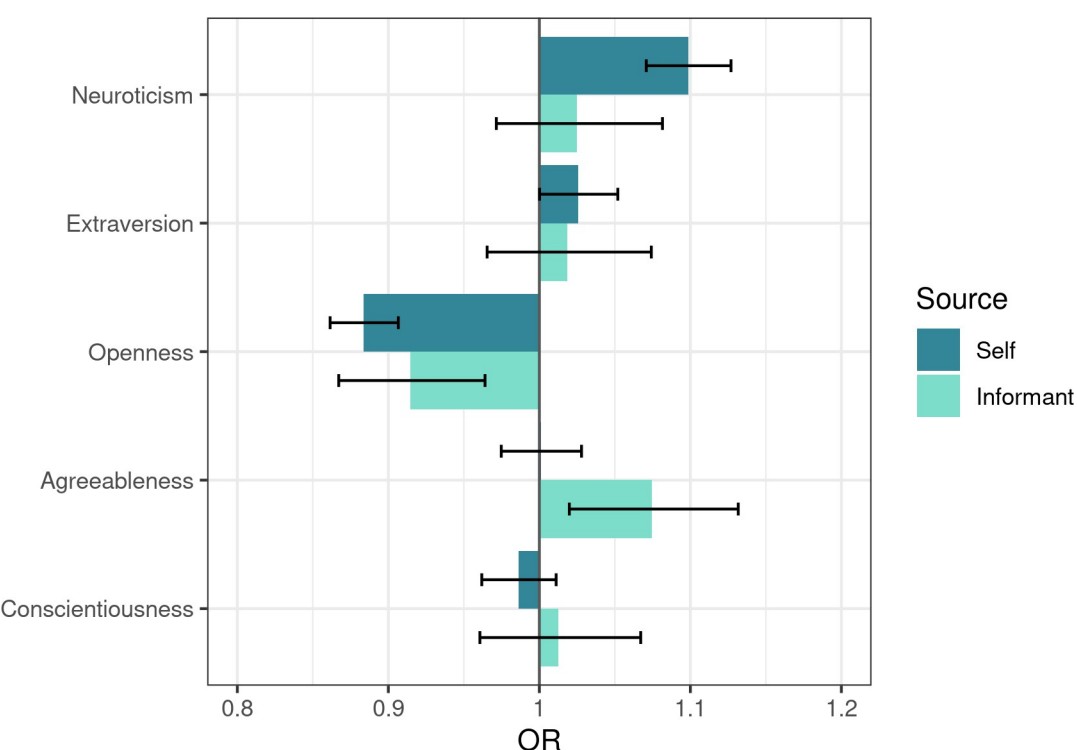

**Fig 2. Five-Factor Model domains' associations with vaccination status.** Associations among all participants. Error bars show 95% confidence intervals. The associations are also shown in S9 Table along with their *p*-values.

no self-reported personality trait predicted vaccination with statistical significance (Fig 4). The self- and informant-rated facets' associations with vaccination status correlated at $r(28) = .34$, $p = .068$.

## Discussion

We aimed to assess whether vaccination against COVID-19 can be predicted from concurrent personality measurements with the broad-coverage 100NP item pool and past personality measurements with the NEO-PI-3. Past measurements of the FFM domains and their facets and items provided limited accuracy in predicting vaccination about 10 years later. Concurrently, the FFM domains also had weak associations with vaccination (with vaccinated people being slightly more agreeable but less open and emotionally stable), whereas the associations were considerably stronger with diverse personality items (with vaccinated people being more science-minded, politically liberal, rule-abiding, and conventional, but less spiritual, bold, and self-assured than unvaccinated people). These associations tended to replicate across self- and informant-reported personality traits. Together, the personality items of the 100NP allowed predicting with 71% accuracy whether people were vaccinated or not after accounting for age, sex, and education; this is considerably higher than either the 50% random-guess baseline or the 62% accuracy provided by five major demographic variables (age, sex, educational status, occupation, and place of residence).

Overall, these results suggest that personality traits' utility in describing and predicting vaccination against COVID-19 may lie in narrow personality traits represented by items, measured not too long before vaccination. The associations with specific traits could reflect some common reasons that people (do not) vaccinate and suggest potential actionable targets for intervention.

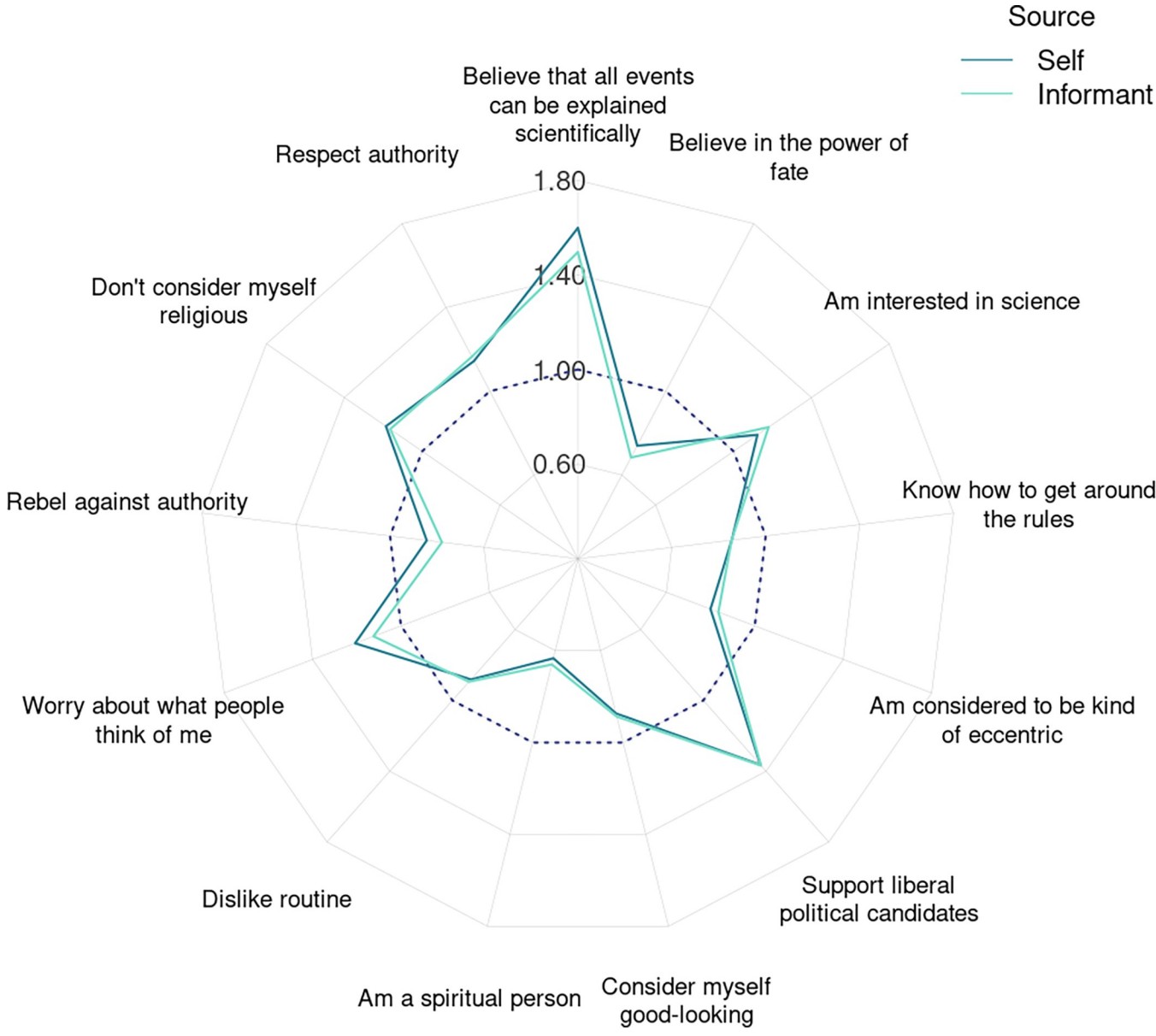

**Fig 3. Associations between vaccination status and personality items.** Associations among all participants. Odds ratios of the items predicting binary vaccination status are shown. The dashed line denotes *OR* = 1.

### Suitable traits for vaccination psychology

To accurately predict whether and when someone is going to vaccinate, it may be most useful to directly ask them about their intentions of doing so. But this does not help us to learn about the proximal psychological factors supporting or hindering vaccination that could be targeted with interventions such as designing vaccination campaigns to accommodate the common hindrances. To identify these, we may study beliefs and attitudes directly related to vaccination in a given societal context (e.g., vaccination is often highly politicized). Indeed, vaccination-related beliefs and attitudes may be able to predict vaccination with higher accuracy than personality traits: In the United States, for example, vaccine acceptance was predicted with a 91% accuracy from a selection of variables concerning either vaccination or its development and

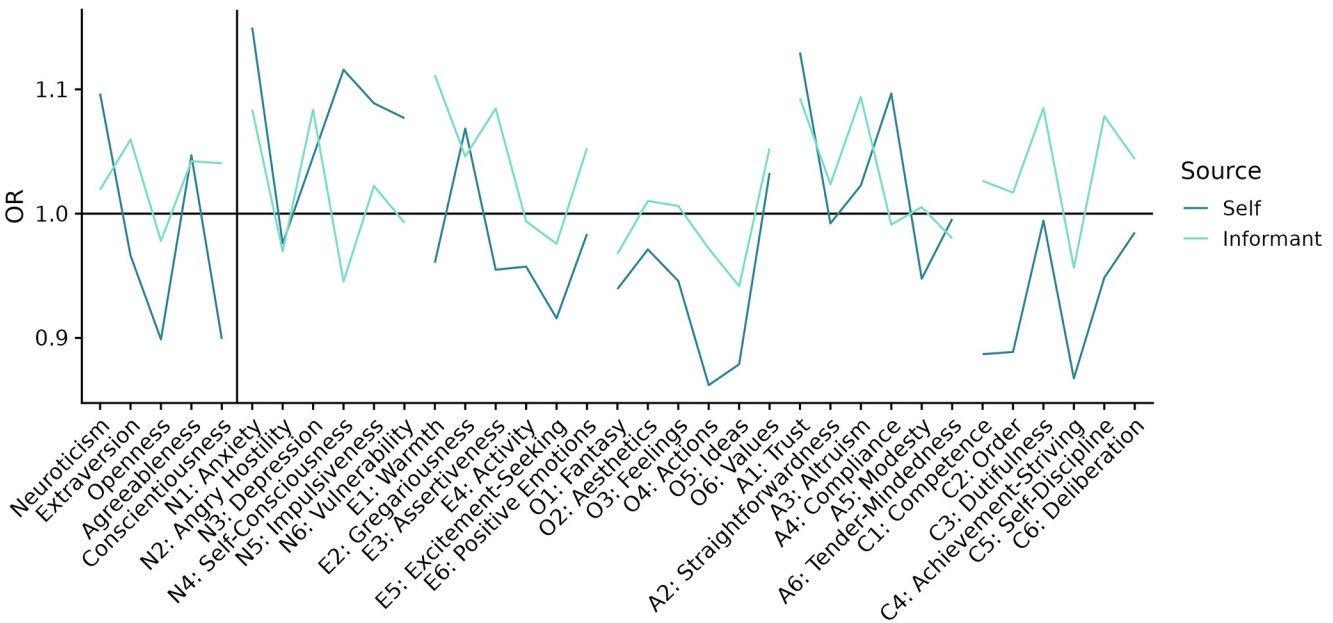

**Fig 4. NEO-PI-3 domains and facets predicting vaccination status.** The associations are reported numerically in S13 Table. None of the correlations reached statistical significance.

delivery, with trust in vaccines being the single most important contributor to the accuracy [56]. Another study predicted vaccine hesitancy with an 82% accuracy from twelve variables including COVID-19-related conspiracy beliefs, paranoia, and mistrust [57]. Although such pandemic- or vaccination-related beliefs and attitudes may be among the most concrete intervention targets to increase vaccine uptake, they may not be easy to change if they are tangled with more general psychological traits. Likewise, knowing that low trust in vaccines predicts vaccine hesitancy still does not tell us much about the psychological reasons for (not) vaccinating. For this, personality traits are more useful.

However, the most commonly used broad personality domains such as the Big Five are only weakly and sometimes inconsistently associated with vaccination. For example, neuroticism has been linked to less and openness to more favourable stances toward vaccination [19, 21, 23, 24, 29], but there have also been reports of openness predicting negative attitudes toward vaccination in general [58]. In our cross-sectional analyses, high neuroticism and low openness weakly predicted vaccination, but these associations were non-significant in our across-time analyses. Whereas non-significant across-time associations can be explained with lower statistical power, a likely explanation for inconsistencies within our own findings and with previous research is the use of different measurement instruments leading to the domains' content differing across analyses.

This issue is avoided in item-level analyses. The narrower personality traits captured by items are not directly related to the pandemic or vaccination, yet they provide non-trivial predictive accuracy for vaccination that is somewhere between the accuracies provided by demographic variables (in our data) and variables pertaining directly to the pandemic and/or vaccination (in previous studies). And what may be at least as important, narrower traits allow researchers less interpretative degrees of freedom than broad domains, thereby reducing the intuition-based just-so storytelling that broad trait constructs can invite. For example, like

some other studies [19–21], we found vaccinated people to be a little more agreeable on average, at least based on informants' ratings. This association, although intuitive, can be explained in any number of ways. The association could be rationalized through agreeableness' compliance facet, as compliant people may find it harder to resist others' expectations that they vaccinate. Or, some people may use anti-vaccination arguments to draw attention to themselves—lack of modesty is also one characteristic of low agreeableness—while yet others may even use these arguments for outright disagreeably picking a good fight. Or, some people may mistrust the intentions of those who implore that they vaccinate—and lack of trust, too, goes with low agreeableness. These all appear plausible just-so explanations, but breaking the domain into narrower traits, like facets or nuances, and linking these with vaccination can constrain such speculations and make them more directly testable.

## Individual personality traits associated with vaccination

Within the agreeableness domain, vaccination had the strongest negative association with the item "Know how to get around the rules." So, some agreeable people may be more willing to get vaccinated primarily because of their compliance—that is, people who report knowing how to get around the rules are likely the same people who can get away without complying with them. For example, at times vaccination was enforced with regulations preventing unvaccinated people from enjoying access to public events and places or travelling. But some people may have been able to work around the restrictions—and these people may have lower agreeableness, on average, possibly contributing to this trait domain's weak correlation with vaccination, especially when "getting around rules" happen to be included in the domain. Meanwhile, other items within the domain, such as modesty and consideration of others, were comparatively less relevant to vaccination. Although prosociality and altruism, for instance, have been associated with self-reported pro-vaccination stances before [21, 22], our results suggest that compliance may be more relevant to objective vaccination behavior.

Among the items associated with neuroticism, vaccination had the strongest positive association with the worry-related ones (e.g., "Am always worried about something"). People who tend to worry may be concerned not only about the pandemic (a possible motivation to get vaccinated) but also vaccines' safety (a possible motivation to refrain from vaccination); the current results suggest that the former may often outweigh the latter, resulting in a net positive population-level association between worry and vaccination. Vaccination was also related to a specific aspect of worry: concern about others' opinions (e.g., "Worry about what people think of me"). Possibly, people who worry about others' opinions are more inclined to comply with the normative expectation to vaccinate as they fear not doing so would draw criticism on them.

People who were rated as bold and self-assured (who "take risks" and are "good at many things"), on the other hand, were more often unvaccinated. Given their apparent confidence, these people may overestimate their knowledge about the virus and vaccines, which could lead to underestimating the associated risks. For instance, a study on knowledge about autism and attitudes toward public vaccination policies showed that overconfidence in one's knowledge regarding the causes of autism was associated with anti-vaccination attitudes but was also highest among people with the least knowledge [59]. Alternatively, as risk-takers, these people may often not feel threatened by the virus. This could be a potential actionable target for vaccination campaigns: Some people may need to be constantly reminded that nobody, including the fearless and healthy, is naturally protected from the virus and can suffer its consequences.

In line with previous results [19, 22, 24, 25], vaccinated people tended to be more politically liberal. However, they also had characteristics that, in contrast to liberalism, are commonly

associated with *lower* openness: On average, vaccinated people were more conventional and more respectful of rules and authority. It may make sense that people who respect rules and follow authority would get vaccinated, given that vaccination is generally advised by government and health authorities. This also fits with unvaccinated people being more eccentric, as we found, and vaccine skeptics seeing and presenting themselves as unconventional, as has been reported previously [58]. Altogether, a disregard for societal conventions or an unwillingness to follow rules may be another factor limiting vaccine uptake.

Most strongly, however, vaccination was related to an evidence-based worldview that may not fit well with the core definitions of any FFM domains. Vaccinated people agreed with statements like "Believe that all events can be explained scientifically" and "Am interested in science." Similarly, vaccination intention is predicted by trust in science [60], but the phrasing of these science-related items we used implies an association with belief in science in principle (not necessarily in practice). Either way, presenting sound evidence on vaccines' safety and effectiveness should remove one potential barrier—uncertainty about the risks and benefits of vaccination—among people with an evidence-based worldview, but such messaging might not be as effective for those least likely to vaccinate because these people may not heed or trust the messaging. Unvaccinated people, on the other hand, more frequently agreed with statements like "Am a spiritual person" and "Believe in the power of fate," consistent with previous evidence linking a spiritual (non-evidence-based) worldview to negative attitudes to vaccination [58, 61]. Also in line with previous reports [62], they were more likely to consider themselves religious. Given that spirituality and religiosity have both been related to negative attitudes toward science [63, 64], empirical evidence may not be enough to motivate spiritual or religious people to vaccinate. Instead, vaccination campaigns could try to normalize vaccination in the context of spiritual beliefs and practices. Spirituality and religiosity, too, do not fit into any FFM domains.

None of these associations were strong. Among other possibilities, this may mean that different traits track vaccination in different subsets of people. For instance, some people could be driven to vaccinate due to their tendency to follow official recommendations, while others may be motivated by their worrying about the pandemic; these motivations do not necessarily apply to the same people. If so, unvaccinated people are not a psychologically homogeneous group, and campaigns targeting any single psychological trait may not be universally successful; instead, it may be reasonable to simultaneously target many of the possible barriers associated with personality traits.

## Subgroup differences: Demographic variables as moderators

Two demographic variables in particular—education and occupation—were consistent moderators in the associations between vaccination and personality traits. People with different levels of education and types of occupations may thus be differently inclined to get vaccinated (partly) due to their personality traits. Because vaccination coverage was higher among people with higher education than those without it and among people in white-collar occupations than those doing blue-collar jobs, we focused on those two groups to identify the traits most relevant to vaccination for people with lower vaccination rates.

Among people without higher education, vaccination was mostly related to the same traits as in the total sample, but in this group, those interested in intellectual activities ("Like to read," "Find political discussions interesting") also had higher odds of being vaccinated. Although the results do not reveal what exactly vaccinated people without higher education liked to read and discuss, intellectual interests may be associated with more deliberation over vaccination and seeking out reliable information, potentially leading to a more balanced perception of its risks and benefits and thus promoting vaccination. Whereas education is

positively related to vaccine literacy [65], meaning that more educated people are better able to find, understand, and use information on vaccines, having intellectual interests may lead to the same outcome even among people without higher education. That is, intellectual inquisitiveness may compensate for lower education in terms of higher vaccination likelihood. An alternative explanation is that people with higher education are more homogeneous in their levels of intellectual interests, in which case the correlation with vaccination would also be lower among them—but still, intellectual interests seem to be relevant at least among people without higher education.

Among those in blue-collar occupations, vaccinated people were more likely to report admitting and apologizing for their mistakes. To the extent that refusing to admit one's mistakes is an indicator of stubbornness more broadly, compared to white-collar workers, unvaccinated blue-collar workers may need more vaccination-favoring evidence to update their beliefs accordingly, especially when they are also confident and have a low tendency to question or rethink their positions. This may be particularly problematic if blue-collar workers have higher risk of being exposed to the virus—that is, it may not be possible for them to work remotely like it is for many white-collar workers. Yet, vaccination was additionally related to items reflecting gregariousness (e.g., "Having good friends is important for me") in this group, suggesting that where other arguments for vaccination fail, social motivations or friends' influences may still be relevant.

## Predicting future vaccination

Personality traits' across-time associations with vaccination status were considerably weaker than concurrent ones. There are several possible reasons for this result. First, the much smaller sample size of the NEO-PI-3 dataset may explain the weaker collective associations as prediction with elastic net benefits from large samples [66]. Second, the NEO-PI-3 sample had a higher average age and, thus, potentially poorer health at the time of vaccination, which could have additionally affected vaccination decisions, increasing the likelihood of vaccination in some and decreasing it in others depending on individual health status. Third, despite people's trait differences being quite stable on average, individuals' personality traits can and do change [67]. For example, even a very high correlation between testing occasions means that about a third of individuals change their trait level substantially within a few years [68]—so trait change within the nearly ten years between personality measurements and vaccination is likely. Hypothesizing that traits contribute to vaccination behavior, concurrent personality trait levels should thus be more strongly linked to vaccination than their past measurements.

A fourth possibility is that the NEO-PI-3 does not capture many of the traits that are most relevant to vaccination, either on the domain or facet level. Item pools like the 100NP that cover a broad range of personality constructs not limited to the FFM domains have an advantage in such multivariable analyses as the diversity of items maximizes the associations. In contrast, the NEO-PI-3 was constructed to measure a group of relatively internally consistent domains and facets [69], which inherently limits their coverage and, thus, predictive utility. It is of course possible that longitudinal models based on NEO-PI-3 items could have allowed predicting vaccination more accurately, but training and validating such models would have required an even larger sample. In sum, although across-time associations may be expected to be weaker than concurrent ones, a larger sample and more diverse item pool could still yield stronger associations.

## Limitations

Interpretations and applications of the results should be mindful of some potential constraints on generalizability. First, whether the results generalize to other populations is unclear as the

associations between specific nuances and vaccination may strongly depend on sociocultural context. Second, because the samples primarily consisted of the ethnic majority (Estonians), results may also not fully generalize to other ethnic groups in Estonia. Third, vaccinated people were somewhat overrepresented in the present study: Vaccination coverage among Estonian adults was 76% as of November 2022 (71% at the beginning of the year), but was 87% and 86% in the 100NP and NEO-PI-3 samples, respectively. This unequal representation suggests that some unvaccinated people were reluctant or otherwise unlikely to become gene donors or participate in the personality studies. Should these people differ in their personality traits from the unvaccinated study participants, this could bias the personality trait–vaccination associations and limit the results' generalizability to all unvaccinated people. In this case, the bottom line is that the reported associations only reflect *some* people's personality-based barriers to vaccination.

Despite the large initial sample sizes, the underrepresentation of unvaccinated people and people vaccinated late, in combination with undersampling, led to small sample sizes in some elastic net analyses, particularly in the NEO-PI-3 dataset. Because elastic net models gain accuracy with larger sample sizes, this likely limited NEO-PI-3 domains' and facets' collective associations with vaccination. Chances are that the domains and facets contain more predictive information than could be detected at the small sample sizes.

Finally, although the 100NP item pool is broad and relatively non-redundant, it is nevertheless not a comprehensive representation of all personality traits. Thus, it may exclude traits relevant to vaccination, also leading to an underestimation of the extent to which personality traits collectively associate with vaccination.

## Conclusion

Objective vaccination status, number of vaccine doses, and vaccination timing can be predicted from both self- and informant-reported narrow personality traits such as those reflecting science-mindedness, liberalism, worry, low self-confidence, and low spiritualism. The prediction accuracy provided by a diverse set of personality items substantially surpasses that of demographic variables. Moreover, the current results suggest that personality traits measured approximately at the time of vaccination may be more informative about vaccination decisions than those measured many years before. Altogether, this knowledge could guide the continued efforts to increase COVID-19 vaccination coverage, as well as inform how vaccines should be communicated to the public in future vaccination programs, in possible future pandemics or otherwise.

## Supporting information

**S1 Table. Sample characteristics: Vaccination details.**
(ODT)

**S2 Table. Informants' characteristics.**
(ODT)

**S3 Table. Items assigned to the five domains in the 100NP.**
(ODT)

**S4 Table. Predicting vaccination from personality traits.**
(ODT)

**S5 Table. Prediction of vaccination from earlier personality measurements with the NEO-PI-3.**
(ODT)

**S6 Table. Prediction accuracies of personality items and demographic variables.**
(ODT)

**S7 Table. Items' elastic net weights.**
(ODT)

**S8 Table. Demographic variables as moderators in the associations between personality traits and vaccination status.**
(ODT)

**S9 Table. Five-Factor Model domains' associations with vaccination status (100NP).**
(ODT)

**S10 Table. Personality items most strongly associated with vaccination status.**
(ODT)

**S11 Table. Personality items most strongly associated with vaccination status among people without higher education.**
(ODT)

**S12 Table. Personality items most strongly associated with vaccination status among people in blue-collar occupations.**
(ODT)

**S13 Table. NEO-PI-3 domains and facets predicting vaccination status.**
(ODT)

**S1 File. Trait-level associations in the demographic groups with lowest vaccination rates.**
(ODT)

## Acknowledgments

The Estonian Biobank Research Team includes the following researchers: Lili Milani (Institute of Genomics, Estonian Genome Centre; lead author for the current study; lili.milani@ut.ee); Andres Metspalu (Institute of Genomics, Estonian Genome Centre); Tõnu Esko (Institute of Genomics, Estonian Genome Centre); Reedik Mägi (Institute of Genomics, Estonian Genome Centre); and Mari Nelis (Institute of Genomics, Estonian Genome Centre).

## Author Contributions

**Conceptualization:** Kadri Arumäe, Tõnu Esko, Krista Fischer, Uku Vainik, René Mõttus.

**Formal analysis:** Kadri Arumäe.

**Funding acquisition:** Tõnu Esko, Uku Vainik.

**Investigation:** Kadri Arumäe.

**Methodology:** Kadri Arumäe, Uku Vainik, René Mõttus.

**Project administration:** Uku Vainik, René Mõttus.

**Resources:** Uku Vainik.

**Software:** Kadri Arumäe.

**Supervision:** Tõnu Esko, Uku Vainik, René Mõttus.

**Visualization:** Kadri Arumäe.

**Writing – original draft:** Kadri Arumäe.

**Writing – review & editing:** Kadri Arumäe, Anu Realo, Liisi Ausmees, Jüri Allik, Tõnu Esko, Krista Fischer, Uku Vainik, René Mõttus.

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
