## [Decision Letter · Decision Letter 0]

16 Aug 2023

PONE-D-23-16332Self- and informant-reported personality traits and vaccination against COVID-19PLOS ONE

Dear Dr. Mõttus,

Thank you for submitting your manuscript to PLOS ONE. After careful consideration, we feel that it has merit but does not fully meet PLOS ONE’s publication criteria as it currently stands. Therefore, we invite you to submit a revised version of the manuscript that addresses the points raised during the review process.

As Reviewer 1 pointed out, some elaboration is necessary in the introduction and discussion sections. On top of that, please provide more explanation for your statement, “Completely separating model training and accuracy testing ensured that predictive accuracies were not overestimated regardless of model complexity.” (rows 297-298). I also wonder why you tested moderation effects by splitting training and testing datasets instead of using more conventional moderation analysis (like interaction effects). Please introduce the “elastic net regression” analysis, as many readers may be unfamiliar with that approach. Why did you choose this method? 

We look forward to receiving your revised manuscript.

Kind regards,

Wojciech Trzebiński, Ph.D.

Academic Editor

PLOS ONE

Journal Requirements:

2. One of the noted authors is a group or consortium [Estonian Biobank Research Team]. In addition to naming the author group, please list the individual authors and affiliations within this group in the acknowledgments section of your manuscript. Please also indicate clearly a lead author for this group along with a contact email address

Reviewers' comments:

Reviewer's Responses to Questions

**Comments to the Author**

1. Is the manuscript technically sound, and do the data support the conclusions?

Reviewer #1: Yes

2. Has the statistical analysis been performed appropriately and rigorously? 

Reviewer #1: Yes

3. Have the authors made all data underlying the findings in their manuscript fully available?

Reviewer #1: No

4. Is the manuscript presented in an intelligible fashion and written in standard English?

Reviewer #1: Yes

5. Review Comments to the Author

Reviewer #1: The work presented is serious, relevant and important in the current state of knowledge. It deals with an important and contemporary issue for the prevention of possible future health crises. The results are important and could benefit the entire scientific community.

I cannot comment on the English level of the article, although it is very comprehensible and accessible to non-native English speakers. The English may need to be checked.

There is a problem with the bibliographical references. The references appear as numbers in brackets in the text, but there are no corresponding numbers in the bibliographic list. Also, sometimes the font size changes.

In the introductory section, I wonder whether the part entitled "Vaccination and the personnality trait hierarchy" should not be brought forward to the beginning of the text to better pose the question and make personality a more important moderator (as the results will show later). Otherwise, the introduction is relevant and interesting. The main reference to the 100NP presentation is missing in the "The Current Study" section, as it is in the "Materials" section. In addition, there should perhaps have been a small review of the literature specific to the 100NP, just as was done on personality concepts.

As for ethical issues, did the participants who completed the questionnaires ten years ago sign a consent form for the data to be used so long after it was collected?

In terms of methodology, why were the data of participants with more than 20% genetic links suppressed? What is the boundary between "with higher education" and "without higher education"?

I'm not in a position to accurately assess the statistical analyses presented in this work. Nevertheless, it seems to me that the authors are not pedagogical enough in this section. The article could be improved by accompanying readers a little better in understanding the choice of analyses and results. Even if they seem to be well conducted, it gives the impression of a lack of transparency.

Please note that in the discussion, having a high score in neuroticism does not mean being neurotic! In fact, researchers now prefer the term "emotional instability" to avoid confusion. Generally speaking, the discussion deserves a summary to get to the point (and use this gain in words to better guide the reader through the results).

Finally, the acknowledgements are not really acknowledgements at all, since they provide access to the study plan.

6. PLOS authors have the option to publish the peer review history of their article (what does this mean?). If published, this will include your full peer review and any attached files.

Reviewer #1: No

---

## [Author Response · Author response to Decision Letter 0]

3 Nov 2023

Revision notes

We thank the Editor and the Reviewer for directing our attention to the areas of our manuscript that needed improvement. We have revised our manuscript with transparency of the methods and analytic choices, as well as clarity and readability, in mind. We address the feedback in more detail below.

Editor: As Reviewer 1 pointed out, some elaboration is necessary in the introduction and discussion sections. On top of that, please provide more explanation for your statement, “Completely separating model training and accuracy testing ensured that predictive accuracies were not overestimated regardless of model complexity.” (rows 297-298). I also wonder why you tested moderation effects by splitting training and testing datasets instead of using more conventional moderation analysis (like interaction effects). Please introduce the “elastic net regression” analysis, as many readers may be unfamiliar with that approach. Why did you choose this method?

Response: Following the Reviewer’s suggestions, we have switched the order of the “Vaccination and the personality trait hierarchy” and “Personality traits and vaccination against COVID-19” sections in the introduction, revising the text for logical order of presentation as necessary. We also made various adjustments in the discussion to provide a more compendious summary and to support overall readability. We clarified the statement regarding separation of training and testing data at lines 313-317: by model complexity, we mean the number of predictors; crucially, the more complex model’s accuracy can be overestimated if training and testing data are not properly separated, which we ensured we did. Separating model training and validation is a standard practice in machine learning and other multi-variable approaches (Mõttus et al., 2020, https://doi.org/10.1002/per.2311; Yarkoni & Westfall, 2017, https://doi.org/10.1177/1745691617693393).

We chose the approach to test moderation effects considering our objective to test whether personality traits together predict vaccination more strongly between the levels of each moderator (e.g., men versus women). Splitting by data by moderator levels allows testing that (Mõttus et al., 2020, https://doi.org/10.1002/per.2311). In contrast, the more standard approach where the predictors’ (personality domains’ or items’) interactions with the moderator are tested would require a separate interaction effect for each predictor. Not only would this be very computationally intensive (especially in the item models due to the large number of predictors) but it would also not yield an estimate on the moderation effect between the predictors together and vaccination. We now clarify the choice of the procedure for testing moderation at lines 357-360 and 363-366 and briefly explain elastic net and its suitability for our objectives at lines 308-313.

Reviewer #1: The work presented is serious, relevant and important in the current state of knowledge. It deals with an important and contemporary issue for the prevention of possible future health crises. The results are important and could benefit the entire scientific community.

I cannot comment on the English level of the article, although it is very comprehensible and accessible to non-native English speakers. The English may need to be checked.

There is a problem with the bibliographical references. The references appear as numbers in brackets in the text, but there are no corresponding numbers in the bibliographic list. Also, sometimes the font size changes.

Response: We have added the missing numbers to the list of references and ensured that the font size is consistent throughout the manuscript.

Reviewer #1: In the introductory section, I wonder whether the part entitled "Vaccination and the personnality trait hierarchy" should not be brought forward to the beginning of the text to better pose the question and make personality a more important moderator (as the results will show later). Otherwise, the introduction is relevant and interesting. 

Response: We have reconsidered the order of the subsections in the introduction and brought the section on the personality trait hierarchy forward.

Reviewer #1: The main reference to the 100NP presentation is missing in the "The Current Study" section, as it is in the "Materials" section. In addition, there should perhaps have been a small review of the literature specific to the 100NP, just as was done on personality concepts.

Response: We have added a reference to the 100NP in the Current Study and Materials sections (lines 159 and 236, respectively). We think that given the current study’s objectives it will suffice to briefly mention the studies that have previously used the 100NP, which we have now also added (lines 250-252). 

Reviewer #1: As for ethical issues, did the participants who completed the questionnaires ten years ago sign a consent form for the data to be used so long after it was collected?

Response: The participants gave their written consent to a form that did not specify any restrictions on the time frame during which their answers could be used. This kind of data use is also in agreement with the Estonian Human Genes Research Act (https://www.riigiteataja.ee/en/eli/508042019001/consolide).

Reviewer #1: In terms of methodology, why were the data of participants with more than 20% genetic links suppressed?

Response: Closely related participants were excluded because high relatedness between particpants can inflate the associations; we now note this at lines 200-201.

Reviewer #1: What is the boundary between "with higher education" and "without higher education"?

Response: As we now also specify in the manuscript (line 299), people were categorized as having higher education if they had obtained at least a bachelor’s or equivalent degree or higher.

Reviewer #1: I'm not in a position to accurately assess the statistical analyses presented in this work. Nevertheless, it seems to me that the authors are not pedagogical enough in this section. The article could be improved by accompanying readers a little better in understanding the choice of analyses and results. Even if they seem to be well conducted, it gives the impression of a lack of transparency.

Response: We made efforts to make the analyses more accessible: we clarified elastic net to the unfamiliar reader (lines 308-313), clarified the reason for choosing this type of a test of moderation (lines 357-360 and 363-366), and made additional small adjustments throughout the statistical analyses section for intelligibility.

Reviewer #1: Please note that in the discussion, having a high score in neuroticism does not mean being neurotic! In fact, researchers now prefer the term "emotional instability" to avoid confusion. Generally speaking, the discussion deserves a summary to get to the point (and use this gain in words to better guide the reader through the results).

Response: We have adjusted our wording and replaced “more neurotic” with “less emotionally stable” (line 512). We have added a clearer summary of the study as a whole to the discussion (lines 521-523) and made some additional adjustments for readability.

Reviewer #1: Finally, the acknowledgements are not really acknowledgements at all, since they provide access to the study plan.

Response: We have removed the links to the study plan and code from the acknowledgements section; they can be found under “Transparency and openness”.

---

## [Decision Letter · Decision Letter 1]

27 Dec 2023

PONE-D-23-16332R1Self- and informant-reported personality traits and vaccination against COVID-19PLOS ONE

Dear Dr. Mõttus,

Thank you for submitting your manuscript to PLOS ONE. After careful consideration, we feel that it has merit but does not fully meet PLOS ONE’s publication criteria as it currently stands. Therefore, we invite you to submit a revised version of the manuscript that addresses the points raised during the review process.

Reviewer 1 accepted your manuscript, but Reviewer 2 raised several thoughts mainly about your data analysis and discussion. Consider those comments to improve your manuscript. Also, please double-check the correctness of the language according to the reviewer's remarks.

We look forward to receiving your revised manuscript.

Kind regards,

Wojciech Trzebiński, Ph.D.

Academic Editor

PLOS ONE

Reviewers' comments:

Reviewer's Responses to Questions

**Comments to the Author**

1. If the authors have adequately addressed your comments raised in a previous round of review and you feel that this manuscript is now acceptable for publication, you may indicate that here to bypass the “Comments to the Author” section, enter your conflict of interest statement in the “Confidential to Editor” section, and submit your "Accept" recommendation.

Reviewer #1: All comments have been addressed

Reviewer #2: (No Response)

2. Is the manuscript technically sound, and do the data support the conclusions?

Reviewer #1: Yes

Reviewer #2: Yes

3. Has the statistical analysis been performed appropriately and rigorously? 

Reviewer #1: I Don't Know

Reviewer #2: Yes

4. Have the authors made all data underlying the findings in their manuscript fully available?

Reviewer #1: (No Response)

Reviewer #2: Yes

5. Is the manuscript presented in an intelligible fashion and written in standard English?

Reviewer #1: Yes

Reviewer #2: Yes

6. Review Comments to the Author

Reviewer #1: (No Response)

Reviewer #2: Self- and Informant-Reported Personality Traits and

Vaccination against COVID-19

PONE-D-23-16332R1

The goal of the present paper was to assess whether Covid-19 vaccination predictions are better (more accurate) when using individual/specific (what the authors describe as a diverse set of narrower traits & facets) rather than general/broader (e.g., Big 5) personality characteristics. The researchers also explored the measurement of these characteristics near to the time of one’s vaccination decision versus temporally further away (i.e., 10 years prior) and compared one’s self reports to the reports of a close other. In sum, I think the results are very interesting and that the paper is promising. However, I do have some concerns and suggestions for improvement and list them below.

More Important Concerns

1. I thought the results were interesting, but did you think about running a hierarchical linear analysis or some form of structural equation modeling, where some factors (predictors) come in early while others come in late? That could potentially boost predictive accuracy and control for potential overlapping variance.

2. I don’t necessarily agree with your comment about liberals following rules more so than conservatives; I think it was overly broad. It could be that both groups follow rules, but it might depend on the context (or one’s ideology). For example, burning down cities and killing other citizens is against the law in most countries, but a couple years ago I saw liberals in the U.S. arguing that it was OK/justified (and that people shouldn’t be put in jail or least not for very long) given it was in protest of systemic racial discrimination. It could also be the case that it depends on the source of the rules/laws. Liberals tend to support rules pushed by the government while conservatives tend to support rules that were established by God.

3. One thing you didn’t spend a lot of time on is trust; obviously some individuals are more trusting of what they see and hear than others. During Covid-19 there was a lot of talk about the “truthfulness” of the info and what constituted misinformation. And public health officials and governments certainly didn’t do themselves any favors in terms of trust when they pushed notions like (my paraphrase) “natural immunity doesn’t work”, “the jab will prevent you from getting covid”, and “people who have gotten the jab won’t transmit covid to others” (or as some claimed “it’s a pandemic of the unvaccinated”), many of which turned out to not be true. But perhaps the vaccinated had more innate trust in others (or institutions) than the unvaccinated.

4. I also took issue with some of the statements you made about blue collar workers. They may be harder to persuade, but unless you can cite evidence to the contrary, I disagree with your notion that (1) they don’t like to admit when they’re wrong or apologize for their mistakes, and (2) they show little tendency to question or rethink their positions. Perhaps it’s better to say that they need more evidence or maybe stronger evidence to update their beliefs (ie, they’re more staunch in their beliefs/set in their ways), whereas white collar workers (who also tend to be more educated and liberal) tend to change their beliefs more often because they have a lower evidence threshold to exceed. I recently saw some neurological evidence that supports this (see, for example, the work of Christman and Jasper). They argue that inconsistent-handers (who have more cross-talk between the right and left brain hemispheres and tend to be politically more liberal) have larger corpus callosa than consistent-handers (who tend to be more conservative).

5. I think the statements you make about application of your findings could be improved. For example, you argue that public health campaigns might be tailored to different sub-groups. It sounds reasonable on the surface, but given that public health campaigns typically target the entire population, how would you pull this off? And if you could pull it off, how would you identify individuals with certain personality characteristics. In another example, you suggest that for the unvaccinated we “normalize” vaccination in the context of spiritual beliefs and practices among religious communities. To some extent, this was tried with Covid-19 (e.g., Billy Graham’s son promoting the vaccine in the U.S.), but it seemed to backfire when people who got the vaccine started dying in droves.

6. Lastly, do you think your results would generalize to other countries, e.g., the U.S.?

Lesser Concerns

1. I noticed there were some places where the grammar could be improved, and as I recall, there were a couple incomplete sentences. But these are easily fixed.

7. PLOS authors have the option to publish the peer review history of their article (what does this mean?). If published, this will include your full peer review and any attached files.

Reviewer #1: No

Reviewer #2: No

---

## [Author Response · Author response to Decision Letter 1]

28 Jan 2024

We thank the Reviewer for suggesting an additional analysis plan and raising important points regarding the interpretation and contextualization of our results. We have made several adjustments to our manuscript which we describe below.

Reviewer 2: The goal of the present paper was to assess whether Covid-19 vaccination predictions are better (more accurate) when using individual/specific (what the authors describe as a diverse set of narrower traits & facets) rather than general/broader (e.g., Big 5) personality characteristics. The researchers also explored the measurement of these characteristics near to the time of one’s vaccination decision versus temporally further away (i.e., 10 years prior) and compared one’s self reports to the reports of a close other. In sum, I think the results are very interesting and that the paper is promising. However, I do have some concerns and suggestions for improvement and list them below.

1. I thought the results were interesting, but did you think about running a hierarchical linear analysis or some form of structural equation modeling, where some factors (predictors) come in early while others come in late? That could potentially boost predictive accuracy and control for potential overlapping variance.

Response: Thank you for the idea of adding predictors at different times. We think this approach would benefit from a solid conceptual basis to determine the appropriate order for the predictors and this more nuanced analysis would be better addressed in a separate paper. For example, should later personality traits be added to models with earlier traits or the other way around? What complicates things further is that the samples have very different sizes, and desirably for cross-validation purposes, only partly overlap, lowering the amount of data the models could capitalize on. Overlapping variance among traits is also addressed in elastic net models which select and shrink predictors. So, we gave this idea considerable thought, but we respectfully decided to not model all assessments in the same model.

Reviewer 2: 2. I don’t necessarily agree with your comment about liberals following rules more so than conservatives; I think it was overly broad. It could be that both groups follow rules, but it might depend on the context (or one’s ideology). For example, burning down cities and killing other citizens is against the law in most countries, but a couple years ago I saw liberals in the U.S. arguing that it was OK/justified (and that people shouldn’t be put in jail or least not for very long) given it was in protest of systemic racial discrimination. It could also be the case that it depends on the source of the rules/laws. Liberals tend to support rules pushed by the government while conservatives tend to support rules that were established by God.

Response: We entirely agree that rule-following could be context-dependent. However, upon reviewing our manuscript, we could not find any instance where we suggest that liberals follow rules more than conservatives. Specifically, in the discussion of liberalism and rule-following, we focus on our results, namely the associations of vaccination with liberalism and rule-following:

 • “Vaccinated people were more likely politically liberal (“Support liberal political candidates,” “Don’t consider myself religious,” “Believe that everyone should have a say,” “Treat all ethnicities and religions equally”), yet more conventional (“Like to be viewed as proper and conventional”), less eccentric (“Am considered to be kind of eccentric”), and more respectful of rules and authority (“Respect authority”) compared to unvaccinated people” (results; lines 478–484)

 • “In line with previous results (46,49,52,54), vaccinated people tended to be more politically liberal. However, they also had characteristics that, in contrast to liberalism, are commonly associated with lower openness: On average, vaccinated people were more conventional and more respectful of rules and authority. It may make sense that people who respect rules and follow authority would get vaccinated, given that vaccination is generally advised by government and health authorities.” (discussion; lines 603–608)

We see how the pattern of associations – people who were by some indicators more liberal were also more likely to vaccinate – may lead to the impression that liberal people are more rule-abiding because vaccination can be seen as abiding to the rules. However, we refrained from making that leap, not least because other evidence quoted above does not support this. As a result, we are grateful to the Reviewer for raising this point, but there is currently nothing we can change in the manuscript.

Reviewer 2: 3. One thing you didn’t spend a lot of time on is trust; obviously some individuals are more trusting of what they see and hear than others. During Covid-19 there was a lot of talk about the “truthfulness” of the info and what constituted misinformation. And public health officials and governments certainly didn’t do themselves any favors in terms of trust when they pushed notions like (my paraphrase) “natural immunity doesn’t work”, “the jab will prevent you from getting covid”, and “people who have gotten the jab won’t transmit covid to others” (or as some claimed “it’s a pandemic of the unvaccinated”), many of which turned out to not be true. But perhaps the vaccinated had more innate trust in others (or institutions) than the unvaccinated.

Response: We thank the Reviewer for this comment. We have briefly discussed the role of trust in science in the context of the correlation between vaccination and evidence-based worldview:

 • “Most strongly, however, vaccination was related to an evidence-based worldview that may not fit well with the core definitions of any FFM domains. Vaccinated people agreed with statements like “Believe that all events can be explained scientifically” and “Am interested in science.” Similarly, vaccination intention is predicted by trust in science (55), but the phrasing of these science-related items we used implies an association with belief in science in principle (not necessarily in practice). Either way, presenting sound evidence on vaccines’ safety and effectiveness should remove one potential barrier—uncertainty about the risks and benefits of vaccination—among people with an evidence-based worldview, but such messaging might not be as effective for those least likely to vaccinate because these people may not heed or trust the messaging.” (lines 612–621)

However, we note that scores on the Trust facet of the NEO-PI-R did not predict vaccination, so the role of general trust in vaccination behavior (as opposed to the rhetoric around vaccination) remains unclear.

Reviewer 2: 4. I also took issue with some of the statements you made about blue collar workers. They may be harder to persuade, but unless you can cite evidence to the contrary, I disagree with your notion that (1) they don’t like to admit when they’re wrong or apologize for their mistakes, and (2) they show little tendency to question or rethink their positions. Perhaps it’s better to say that they need more evidence or maybe stronger evidence to update their beliefs (ie, they’re more staunch in their beliefs/set in their ways), whereas white collar workers (who also tend to be more educated and liberal) tend to change their beliefs more often because they have a lower evidence threshold to exceed. I recently saw some neurological evidence that supports this (see, for example, the work of Christman and Jasper). They argue that inconsistent-handers (who have more cross-talk between the right and left brain hemispheres and tend to be politically more liberal) have larger corpus callosa than consistent-handers (who tend to be more conservative).

Response: We should clarify that while vaccinated blue-collars reported more readily admitting and apologizing for their mistakes according to our results, we do not suggest that blue-collar workers do not like to admit being wrong, refuse to apologize for their mistakes, or show little tendency to question or rethink their positions. However, to avoid leaving a wrong impression, we now have replaced our statement about blue-collar workers being hard to persuade with their potentially needing more or stronger evidence. The revised discussion section on blue-collar workers reads:

 • “Among those in blue-collar occupations, vaccinated people were more likely to report admitting and apologizing for their mistakes. To the extent that refusing to admit one’s mistakes is an indicator of stubbornness more broadly, compared to white-collar workers, unvaccinated blue-collar workers may need more vaccination-favoring evidence to update their beliefs accordingly, especially when they are also confident and have a low tendency to question or rethink their positions. This may be particularly problematic if blue-collar workers have higher risk of being exposed to the virus—that is, it may not be possible for them to work remotely like it is for many white-collar workers.” (lines 659–668)

Reviewer 2: 5. I think the statements you make about application of your findings could be improved. For example, you argue that public health campaigns might be tailored to different sub-groups. It sounds reasonable on the surface, but given that public health campaigns typically target the entire population, how would you pull this off? And if you could pull it off, how would you identify individuals with certain personality characteristics. In another example, you suggest that for the unvaccinated we “normalize” vaccination in the context of spiritual beliefs and practices among religious communities. To some extent, this was tried with Covid-19 (e.g., Billy Graham’s son promoting the vaccine in the U.S.), but it seemed to backfire when people who got the vaccine started dying in droves.

Response: We clarify that rather than identifying individuals with certain personality characteristics (which would indeed require additional effort), campaigns could be designed to target the major reasons for non-vaccination, combining knowledge of relevant traits in demographic subgroups with vaccination rates in those subgroups. We made this clearer in the introduction:

 • “While some personality trait–vaccination relations may turn out to generalize across individuals’ demographic backgrounds, other traits could be more relevant for vaccination decisions in some groups than others. There is already evidence that personality traits’ associations with vaccination willingness may depend on age and sex. For instance, traits reflecting risk aversion and self-interest could drive older people to vaccinate, while prosocial traits could be more relevant for young people, particularly men (19). If such moderation effects of age, sex, or other demographic variables exist, knowing them could provide further information about the specific drivers of and hindrances to vaccination, potentially allowing for more targeted interventions. For instance, if more anxious older people are apprehensive of vaccines, then campaigns, especially those targeted at this demographic group, could focus on vaccines’ safety or risk-benefit trade-offs; if younger men’s decision to vaccinate is linked to prosociality, campaigns could appeal to altruistic motives, especially if targeted at this demographic group. It would be especially valuable to understand what personality traits are most strongly related to vaccination in groups with the lowest vaccination uptake because effectively motivating these people could have the highest impact on overall vaccination rates. Vaccination campaigns could thus prioritize messages most likely to be effective in subgroups with the lowest vaccine uptake.” (lines 112–127)

Reviewer 2: 6. Lastly, do you think your results would generalize to other countries, e.g., the U.S.?

Response: We have revised our discussion of generalizability to enhance its clarity. The beginning of our limitations section now reads as follows:

 • “Interpretations and applications of the results should be mindful of some potential constraints on generalizability. First, whether the results generalize to other populations is unclear as the associations between specific nuances and vaccination may strongly depend on sociocultural context. Second, because the samples primarily consisted of the ethnic majority (Estonians), results may also not fully generalize to other ethnic groups in Estonia. Third, vaccinated people were somewhat overrepresented in the present study: Vaccination coverage among Estonian adults was 76% as of November 2022 (71% at the beginning of the year), but was 87% and 86% in the 100NP and NEO-PI-3 samples, respectively. This unequal representation suggests that some unvaccinated people were reluctant or otherwise unlikely to become gene donors or participate in the personality studies. Should these people differ in their personality traits from the unvaccinated study participants, this could bias the personality trait–vaccination associations and limit the results’ generalizability to all unvaccinated people. In this case, the bottom line is that the reported associations only reflect some people’s personality-based barriers to vaccination.” (lines 694–706)

Reviewer 2: 1. I noticed there were some places where the grammar could be improved, and as I recall, there were a couple incomplete sentences. But these are easily fixed.

Response: We have reread the manuscript and made adjustments to improve the language.

---

## [Decision Letter · Decision Letter 2]

20 Feb 2024

Self- and informant-reported personality traits and vaccination against COVID-19

PONE-D-23-16332R2

Dear Dr. Mõttus,

We’re pleased to inform you that your manuscript has been judged scientifically suitable for publication and will be formally accepted for publication once it meets all outstanding technical requirements.

Kind regards,

Wojciech Trzebiński, Ph.D.

Academic Editor

PLOS ONE

Additional Editor Comments (optional):

Reviewers' comments:

Reviewer's Responses to Questions

**Comments to the Author**

1. If the authors have adequately addressed your comments raised in a previous round of review and you feel that this manuscript is now acceptable for publication, you may indicate that here to bypass the “Comments to the Author” section, enter your conflict of interest statement in the “Confidential to Editor” section, and submit your "Accept" recommendation.

Reviewer #2: All comments have been addressed

2. Is the manuscript technically sound, and do the data support the conclusions?

Reviewer #2: (No Response)

3. Has the statistical analysis been performed appropriately and rigorously? 

Reviewer #2: (No Response)

4. Have the authors made all data underlying the findings in their manuscript fully available?

Reviewer #2: (No Response)

5. Is the manuscript presented in an intelligible fashion and written in standard English?

Reviewer #2: (No Response)

6. Review Comments to the Author

Reviewer #2: (No Response)

7. PLOS authors have the option to publish the peer review history of their article (what does this mean?). If published, this will include your full peer review and any attached files.

Reviewer #2: No

---

## [Editor Report · Acceptance letter]

4 Mar 2024

PONE-D-23-16332R2 

PLOS ONE

Dear Dr. Mõttus, 

I'm pleased to inform you that your manuscript has been deemed suitable for publication in PLOS ONE. Congratulations! Your manuscript is now being handed over to our production team.

Kind regards, 

on behalf of

Dr. Wojciech Trzebiński 

Academic Editor

PLOS ONE